

# Temporal and spatial variability of terrestrial diatoms at the catchment scale: controls on productivity and comparison with other soil algae

Jasper Foets[1,2], Carlos E. Wetzel[1], Adriaan J. Teuling[2] and Laurent Pfister[1,3]

[1] Environmental Research and Innovation Department, Luxembourg Institute of Science and Technology, Belvaux, Luxembourg
[2] Department of Environmental Sciences, Wageningen University and Research, Wageningen, Netherlands
[3] Faculty of Science, Technology and Medicine, University of Luxembourg, Belval, Luxembourg

## ABSTRACT

Terrestrial diatoms are an integral component of the soil microbial community. However, their productivity and how it compares to other algal groups remains poorly known. This lack of knowledge hampers their potential use as environmental markers in various applications. As a way forward, we investigated the seasonal and spatial patterns of diatom assemblages and the role of environmental factors on the soil diatom productivity. We collected soil algal samples in 16 sites across the Attert River basin (Luxembourg) every 4 weeks for a period of 12 months. The algal abundances were then derived from pigment analysis using High-Performance Liquid Chromatography. Our results indicate that diatom productivity is mainly controlled by factors related to soil moisture availability leading to seasonal patterns, whereas the concentration of green algae remained stable over the course of the study period. Generally, anthropic disturbed habitats contained less living diatom cells than undisturbed habitats. Also, we learned that diatoms can be the dominant algal group at periods of the year with high soil moisture.

# INTRODUCTION

Diatoms are microscopic, unicellular algae and form one of the most common and diverse algal groups in both freshwaters and marine environments (*Round, Crawford & Mann, 1990*). They are pigmented, photosynthetic and because of their high abundances, they play a large role in the exchange of gasses between the atmosphere and biosphere. It has been estimated that they are responsible for 20% of the total oxygen production on the planet (*Scarsini et al., 2019*). While diatoms are generally regarded as inhabitants of water bodies, numerous taxa are able to survive and reproduce in a variety of terrestrial ecosystems such as soils, mosses, wet walls and rocks (*Smol & Stoermer, 2010*). Generally, those environments are much harsher for diatoms than aquatic habitats (*Ress, 2012*).

Corresponding author
Jasper Foets, jasper.foets@list.lu

Variables such as moisture and temperature can vary considerably not only seasonally, but also over the course of a day or between 2 consecutive days. As a consequence, diatoms are typically exposed to frequent and prolonged periods of desiccation. Terrestrial diatom species have developed several features (both morphologically and physiologically) to cope with the temperature variability and limited moisture conditions. For instance, as the characteristic siliceous cell wall consists of many pores, they often decrease its number or create structures to enclose them on the out-or inside to prevent water loss (*Lowe et al., 2007*; *Ress, 2012*). This adaptability suggests that diatoms do not only survive on soils, but that they eventually also could thrive and even be the dominant algal group in a terrestrial environment.

As an integral component of the soil microbial community, diatoms play an important role in the functioning of the soil ecosystem. By moving on or being attached to the soil surface, they produce an extracellular matrix of mucopolysaccharides, which will bind soil particles and eventually stabilize the soil (*Booth, 1941*; *Paterson, 1986*; *Tolhurst, Gust & Paterson, 2002*; *Jewson, Lowry & Bowen, 2006*). This aggregation will subsequently reduce water loss by evaporation, limit soil erosion and improve water infiltration, thereby providing a favorable habitat for seed germination (*Booth, 1941*; *Hoffmann, 1989*). Moreover, they promote the release of nutrients from insoluble compounds and the weathering of silicates by creating a slightly acidic environment (*Hoffmann, 1989*; *Wu et al., 2013*). *Schmidt, Dyckmans & Schrader, 2016* found that they even might be an important carbon input source for decomposing soil animals. Their death and decay also provides organic matter, which other microorganisms and plants can potentially utilize (*Fritsch, 1907*; *Shubert & Starks, 1979*; *Starks, Shubert & Trainor, 1981*). As climate change is highly likely to increase the frequency and persistence of droughts in the near future (*Berg & Sheffield, 2018*), there is growing concern about potential negative impacts that this change could have on the viability and functions of terrestrial diatoms and subsequently also on higher plants.

In temperate regions, the seasonal variability in meteorological (e.g., humidity), chemical (e.g., nutrients) or biological conditions (e.g., vegetation height) is not causing any significant changes in the composition of soil diatom communities according to *Lund (1945)* and *Foets et al. (2020)*. However, seasonal variations might prevail in the diatom biomass with possible maxima observed in spring and autumn in response to changes in temperature and precipitation (*Lund, 1945*). A similar observation for soil algae was reported by *Stokes (1940)*, who noticed sharp increases in algal numbers after periods of rain and that the optimum moisture range for algal growth lies between 40% and 60% of the moisture-holding capacity of the soil. This amount would probably be ideal to produce a soil solution with enough available nutrients to support algal growth (*Pringsheim, 1950*; *Starks & Shubert, 1982*). Furthermore, *Davey (1991)* and *Grondin & Johansen (1995)* reported an increased algal biomass (in algal numbers and in chlorophyll a concentration) after snow cover. Although *Lund (1945)* is the only study (of all aforementioned studies) to date to have analyzed specifically the terrestrial diatom production, the temporal and spatial scale of that study was relatively limited (i.e., one garden soil in optimal conditions). There is a pressing need for a better understanding of
the seasonal patterns in diatom production across contrasted environmental settings. This stands as a prerequisite for their subsequent use in future applications (e.g., as soil quality bio-indicator or as a hydrological tracer).

Algae and cyanobacteria use pigments and other compounds to regulate the spectral composition and the intensity of incoming light (photon flux density). This allows them to maximize photosynthetic efficiency whilst preventing photochemical degradation of cellular components and indirect loss of function mediated by reactive oxygen species (*Vincent, 2000*). Two types of pigments are involved in those processes. Chlorophylls, particularly a, b and c, are the central components in light-harvesting, while carotenoids, including xanthophylls, are mainly engaged in photoprotection (*Demmig-Adams & Adams, 2000*). In contrast to higher plants, which all contain the same major carotenoids, algae have different characteristic carotenoids. So far, around 1,100 different natural carotenoids have been characterized and reported in the literature (*Fernandes et al., 2018*). This diversity and specificity allows us to use certain carotenoids as so-called biomarkers or chemotaxonomical markers to infer the production of different algal groups. Furthermore, recent developments in High-Performance Liquid Chromatography (HPLC) allow for rapid and accurate determinations of pigments and their derivatives in aquatic and sediment samples. Pigment concentrations have now become a common tool in paleolimnological research and water quality assessments (*Verleyen et al., 2004*; *Reavie et al., 2017*; *Duong et al., 2019*). However, except for a few studies on soil primary production (*Shubert & Starks, 1979*; *Starks & Shubert, 1982*; *Davey, 1991*; *Davey & Rothery, 1992*), no other attempts have been reported to date to assess soil diatom production and its environmental controls.

Previously, we have investigated the temporal and spatial variability in the composition of soil diatom communities (*Foets et al., 2020*). There, we found that forests create a stable microhabitat for diatoms, that the temporal variation of the communities is mainly controlled by farming practices, and that they need 1 to 2 months to reestablish a new, stable community after a significant change in the environment. Here, we leverage these findings and investigate how diatom production changes over space and time in a mesoscale catchment compared to green algae and cyanobacteria. The catchment scale was selected, since this is the natural unit in which environmental variables such as radiation, temperature and soil moisture show the largest variability. The aim of this study is mainly exploratory, but while analyzing the data for *Foets et al. (2020)*, we also developed two hypotheses. Our first hypothesis states that soil diatom productivity (i.e., concentration of fucoxanthin) is controlled by meteorological conditions. Our second hypothesis stipulates that the biomass changes depending on the type of habitat, since communities in (anthropic) disturbed areas contain generally larger species (i.e., higher pigment concentration) compared to undisturbed areas (*Foets et al., 2020*).

## MATERIALS AND METHODS

### Study area and weather conditions

Our study site is located in the Attert River basin in Luxembourg, covering an area of approximately 249 km$^2$ (Fig. 1). The western limit of the basin extends into Belgium

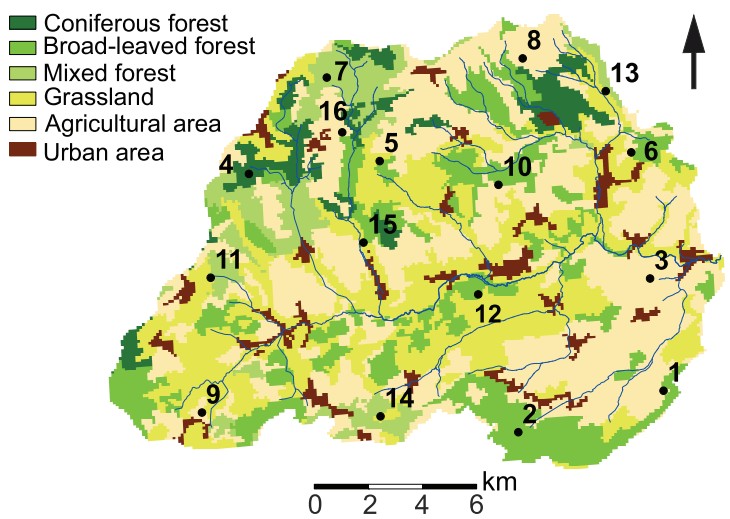

**Figure 1 Land use characteristics of the Attert River basin in Luxembourg and Belgium.** The map shows the 16 sampling locations. Source: modified from *Corine Land Cover (2018)*.

(49°46′13.0″ N, 5°59′9.2″ E). Additional information on the study area is given in *Foets et al. (2020)*.

We took monthly samples from December 2017 to November 2018. The summer season prior to the start of our sampling campaign had been relatively wet (on average 88 mm/month; that is, 32 mm/month more than in reference period 1981–2010), whereas the average monthly precipitation during the sampling period was 56.8 ± 33.5 mm. The maximum monthly precipitation was observed in December 2017 and January 2018 (111.9 and 123.5 mm, respectively). Monthly precipitation reached a minimum in October 2018 (18 mm). July and August were the warmest months, reaching mean monthly temperatures of 20.5 °C and 18.7 °C, respectively. Most freezing days occurred in February and March 2018 (28 and 12, respectively) (Data obtained from the "Administration des Services Techniques de l'Agriculture" (ASTA)). Overall, 2018 was characterized by a cold winter and an exceptionally dry and warm summer period (*Meteolux, 2019*).

## Meteorological data

Daily meteorological data for the entire study period was retrieved from two weather stations (Useldange and Roodt) situated in the river basin. All data was taken from Useldange, except daily precipitation. For that, we calculated Thiessen polygons to know which sites were closer to the station. Besides, we derived air temperature for every site following the general equation: −0.65 °C × 100 m (Altitude). The station in Useldange is located at 280 m a.s.l. Additionally, 2 °C were deducted for the forested sites. Finally, the 7-day average prior to the sampling day was taken for every meteorological variable.

## Soil sampling and pigment analysis

In order to capture the intra-annual and spatial variability of the absolute algal abundances, we took soil samples at the soil surface in 16 sites distributed across the Attert River basin (*Foets et al., 2020*). The sampling campaigns took place every 3 to 5 weeks for a period of 12 months (*n* = 192 samples). During each campaign, we made a description of the site in order to keep track of any potential changes in the environment in between the sampling periods. Characteristics such as type of land use, disturbances, height of the lower vegetation (we used grass height as an indicator for mowing/grazing) were noted (see Table A1). In addition, we measured soil moisture content and electrical conductivity (EC) 30 times, using a FieldScout TDR 300 Soil Moisture Meter. Moreover, out of the four soil samples that we took per site, one was to serve for pigment analysis and one was used for species community analysis (*Foets et al., 2020*).

We used metal cylinders (Ø: 5.6 cm, height: 4 cm) to collect small soil cores. Upon arrival in the lab, all plant litter was carefully removed from the top soil layer to keep the sample undisturbed. We then extracted algae by rinsing the superficial layer with Milli-Q water until a 50 mL falcon tube was completely filled (*Barragán, Wetzel & Ector, 2018*). The samples, consisting of soil particles and water, were kept in the dark at 4 °C until pigment extraction.

The following day, we centrifuged the samples for 10 min at 4,000 rpm. We discarded the supernatants and freeze-dried the residues for approximately 24 h. Next, we extracted the pigments two times with five mL methanol by shaking for 15 min at 25 Hz. with a mixer mill (MM 400 Retsch). After centrifugation (4,000 rpm., 10 min), organic phases were pooled and extracts were concentrated by vacuum evaporation.

We separated and quantified the pigments by HPLC with Diode-Array Detection (HPLC-DAD, 1260 Infinity; Agilent, Santa Clara, CA, USA). An Acquity UPLC HSS T3 column at 30 °C was used (2.1 × 100 mm, 1.7 μm) with a mobile phase consisting of 50 mM ammonium acetate solution, acetonitrile/dichloromethane/methanol (75/10/15 v/v/v) and ethyl acetate. Pigments were eluted at a flow rate of 300 μL min$^{-1}$ with a ternary gradient. We identified the compounds of interest by their retention time and compared the spectral data with certified standards provided by Sigma–Aldrich and Carotenature. The detection limit for the standards was 30 ng per sample. Quantification was done at 450, 464, 620 and 655 nm using external calibration. We expressed chlorophyl and carotenoid concentrations in ng per sample (50 mL) and transformed them to μg L$^{-1}$ to make the concentrations comparable with other studies. The different pigments used in the analysis, their affinity and interpretation are listed in Table 1.

## Physico-chemical analysis

After the algal extraction, we only kept the first cm of soil of each of the four samples. After having dried the samples for one week at room temperature, we crushed the soils and sieved them at 2 mm. We measured the pH according to the *ISO 10390 (2005)* standard. The bioavailable part of nutrients (phosphorus (P), Total Nitrogen (TN), Dissolved Organic Carbon (DOC), silica (Si), magnesium (Mg), manganese (Mn), sodium (Na), potassium (K), iron (Fe) and aluminum (Al)) were all extracted following the method described by

**Table 1 Summary of proxies employed in this study.**

| Pigment | Code | Affinity | Interpretation |
|---------|------|----------|----------------|
| Chlorophyll a (incl. derivatives) | TChla | All photosynthetic algae | Primary production/total algal production |
| Fucoxanthin | Fx | B[*] | Diatom production |
| Diatoxanthin | Da | B, X[*], E | Diatoms, measure for irradiance |
| Diadinoxanthin | Dd | B, X[*], E[*] | Diatoms, measure for irradiance |
| Zeaxanthin | Ze | Cy[*], R[*], B, E, Cl | Cyanobacteria production |
| Lutein | Lu | Cl[*] | Green algal production |
| (Dd +Da) Fx$^{-1}$ | | | Mean irradiance per diatom |

Notes:
[*] Major carotenoid in most species of the class.
Cy, Cyanophyta (cyanobacteria); E, Euglenophyta; CL, Chlorophyta (~green algae); B, Bacillariophyceae (~diatoms); R, Rhodophyta; X, Xanthophyceae. Table derived from *Verleyen et al. (2004)* and *Takaichi (2011)*.

*Houba et al. (2000)*. We added 150 mL of 0.01M $CaCl_2$ to 15 g of soil in a 250 mL glass bottle and shook this mixture for 2 h. Next, we centrifuged the samples for 15 min at 5,000 rpm. The supernatant was filtered through 0.7 μm glass microfiber filters and stored at 4 °C prior to the ICP-OES (5110 VDV radial; Agilent, Santa Clara, CA, USA) analysis of P, Mg, Mn, K, Si and Na. DOC and TN concentrations were determined directly after the extraction with a Torch Combustion TOC/TN analyzer (Teledyne Tekmar, Mason, OH, USA). With a view to Fe and Al content analysis, we additionally filtered 10 ml of the supernatant with 0.45 μm syringe filters (Acrodisc®; Pall, New York, NY, USA) and added 100 μL 1M HCl before the ICP-MS (Thermo Elemental X7/Perkin-Elmer© DRC-e) analysis. We did not estimate the free metal ions concentrations.

## Scanning electron microscope analysis

We carried out a scanning electron analysis to get information regarding diatoms within the soil matrix (e.g., their abundance, their way of living). Additional soil samples were taken in November 2017 at an agricultural grassland site (site 9, see *Foets et al., 2020*). We carefully placed a part of the soil samples on aluminum stubs and analyzed them with a Quanta 200 Field Emission Gun Scanning Electron Microscope (FEG SEM) (Phillips-FEI). In addition, the microscope was equipped with a Genesis XM 4i Energy Dispersive Spectrometer (EDS) system for chemical analysis. For complementary EDS analysis, we used a large field detector, as our examination was done in a low-vacuum environment (150 Pa).

## Statistical analyses

Based on meteorological and discharge datasets, we calculated two soil moisture availability proxies for the Attert River basin. First, we inferred daily water storage deficits (SD) from water balance calculations as per *Pfister et al. (2017)*. Second, we calculated daily differences between precipitation input and potential evapotranspiration loss (PET). We used the FAO Penman-Monteith equation for calculating PET (*Allen et al., 1998*).

For these calculations, we assumed the soil heat flux to be zero and the psychrometric constant equal to 0.065.

Correlations between continuous environmental variables and between pigment concentrations were assessed using Spearman rank correlation. One of each correlated (i.e., $R^2 > 0.5$ or $< -0.5$) environmental variable was retained for further discriminant analyses (see "Results" section). Also, possible spatial autocorrelation of the pigment measurements was assessed with a Mantel test. For doing that, pigment data were transformed to a dissimilarity matrix (Bray–Curtis) and tested if there was significant correlation (perm-999) with the geographical distance matrix. The latter was obtained using the *rdist.earth* function from the R-package *fields* (*Nychka et al., 2017*). Before carrying out the indirect and direct ordination analysis, pigment data was Hellinger-transformed. Detrended correspondence analysis (DCA) on the pigment data revealed a gradient length smaller than 2.0 S.D. (standard deviation) (*Hill & Gauch, 1980*). Therefore, we relied on a redundancy analysis (RDA) as exploratory method.

We used the variation partitioning technique (*Legendre, Borcard & Peres-Neto, 2005*) to quantify the relative contributions of three matrices of variables (soil chemistry, meteorology and site characteristics) to the explained variance and to test whether and to what extent their contributions to the composition of the pigment concentrations could be separated. Testable fractions were analyzed with RDA. For testing our first hypothesis, we carried out a multiple regression analysis on the fucoxanthin data to identify the variables that are significantly related to diatom abundances. A generalized mixed model (*lmer* function) was formed with "site" as a random variable to account for repeated measurements. Prior to model building, fucoxanthin concentrations were log-transformed (x + 0.0001) to get normal distributed data and environmental variables were rescaled. We did variable selection with backward elimination using the *step* function from the *lmerTest* package (*Kuznetsova, Brockhoff & Christensen, 2017*). Afterwards, the residuals of the final model were checked. For testing our second hypothesis, we calculated diatom biovolume by multiplying the raw diatom community data given in *Foets et al. (2020)* with the species size measurements given in *Rimet & Bouchez (2012)* and Omnidia (version 6.0.8, 2018) (*Lecointe, Coste & Prygiel, 1993*). Next, we tested the variability of biovolume and pigment concentrations over qualitative variables (e.g., month, soil and habitat type) with analysis of variance (ANOVA), whilst accounting for repeated measurements, or the non-parametric Kruskal–Wallis test. Homoscedasticity was checked with the Breusch–Pagan test. Thereafter, we analyzed more specifically the significant results by relying on Tukey's range or the pairwise Wilcoxon test, respectively. All aforementioned statistical analyses were performed using the R statistical program (R v. 3.5.3.; http://www.r-project.org/) and additional functions from the R-package *vegan* version 2.5-5 (*Oksanen et al., 2019*).

## RESULTS

### Soil matrix

Diatoms live at the top of the soil surface and with 15 cells per approximately 0.2 mm$^2$ (= 7,500 cells cm$^{-2}$) diatoms are relatively abundant even on a small area such as

shown in Fig. 2B. The occurring species are mainly *Nitzschia pusilla* (Kützing) Grunow emend. Lange-Bertalot, *Sellaphora nana* (Hustedt) Lange-Bertalot, Cavacini, Tagliaventi & Alfinito and *Hantzschia amphioxys* (Ehrenberg) Grunow. Bacilli-shaped bacteria are also present at the soil surface (Fig. 2C). However, because they are so small, it is impossible to see whether the bacteria are cyanobacteria or not. Green algae are absent on the picture. This general observation is supported by the average pigment concentrations measured on that particular site (site 9), which indicate that diatoms on average (9.0 μg L$^{-1}$) are more abundant than green algae (2.6 μg L$^{-1}$) and cyanobacteria (0.7 μg L$^{-1}$).

Besides the topography, the SEM picture also gives an idea of the chemical composition of the top layer (Table 2). The contrast of the picture is based on the convention that "heavier" elemental areas (= higher atomic number) are shown in light gray and low atomic number elements in darker gray. Our subsequent analysis showed that the percentage of carbon and oxygen in *H. amphioxys* is much higher than at the soil surface (difference of respectively ± 100 and 50%), while the silica concentration is approximately the same. Interestingly, the aluminum concentration at the soil is high (11.0%) compared to the other elements and around a third of that percentage is found in the diatom (3.4%). In addition to the composition mode, chemical maps could be made displaying the chemical contrast in a colored pixel map (Figs. 2G–2I). This means that the higher the elemental concentration is, the higher the colored pixel density will be. Since oxygen and silica had the highest concentrations, they gave the best pixel maps of all present chemical elements. While the map of silica is only able to show us roughly the contours of *H. amphioxys*, some differences in the external valve structure are visible on the oxygen map.

## Controlling environmental variables

Chlorophyll a (cf. primary production (Table 1)) had the highest average concentration over the sampling campaign (12.7 ± 13.5 μg L$^{-1}$), followed by lutein (cf. green algae) and fucoxanthin (cf. diatoms), with concentrations of 5.7 ± 10.0 μg L$^{-1}$ and 5.0 ± 8.2 μg L$^{-1}$, respectively. We determined the lowest average concentrations in zeaxanthin (cf. cyanobacteria; 1.1 ± 1.5 μg L$^{-1}$) and diatoxanthin (0.6 ± 0.3 μg L$^{-1}$). However, the differences between the concentrations of fucoxanthin and lutein were not significant when considering the whole dataset (Wilcoxon test, $P = 0.089$), whereas the concentrations of zeaxanthin were significantly different for both fucoxanthin and lutein ($P < 0.001$). This finding is confirmed when comparing the different habitat types with each other (Fig. 3A). In addition, the figure shows that the type of habitat does not have an influence on those three algal groups (Kruskal–Wallis, df = 4, $P > 0.05$). However, diatom biovolume significantly differs between the different habitats with disturbed areas having generally larger communities than undisturbed habitats (ANOVA, df = 4, $P < 0.001$) (Fig. 3B).

Results of the correlation analyses showed that the environmental descriptors, physico-chemical and meteorological variables were not correlated between each other ($R^2$ not higher than 0.5 for all the variables). DOC and TN ($R^2 = 0.58$) and EC and moisture ($R^2 = 0.91$) were correlated. Furthermore, most of the meteorological variables were correlated with each other ($R^2 > 0.5$ or $< -0.5$) and as a result, only radiation and SD

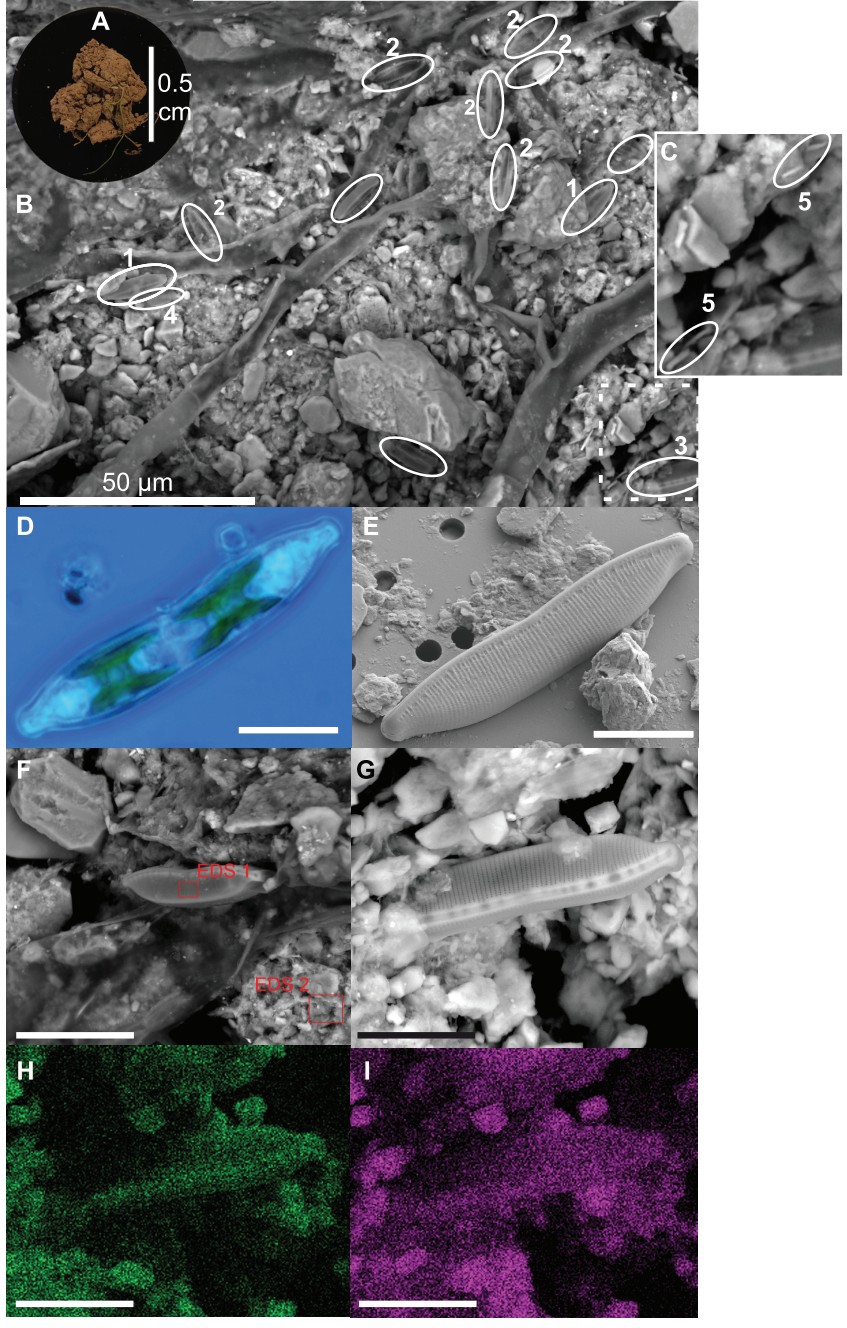

**Figure 2 Illustration of diatoms living on the soil surface.** Images were taken with a Scanning Electron Microscope (SEM). The soil sample is from an agricultural grassland site (site 9, see Fig. 4 in *Foets et al. (2020)*) sampled in November 2017. (A) Soil sample on aluminum stub. (B) SEM image. (C) Close-up of the dashed rectangle showing bacilli-shaped bacteria. (D) Life material showing pigment distribution in *Hantzschia amphioxys* (Ehrenberg) Grunow. (E) SEM picture showing external valve structures. (F) SEM picture showing the EDS 1 and EDS 2 analyzed zones. (G) SEM picture. (H) Chemical map showing oxygen concentration at the surface. (I) Chemical map showing silica concentration at the surface. (1) *Sellaphora nana* (Hustedt) Lange-Bertalot, Cavacini, Tagliaventi & Alfinito; (2) *Nitzschia pusilla* (Kützing) Grunow emend. Lange-Bertalot; (3) *H. amphioxys*; (4) *Mayamaea* sp.; (5) bacteria. The scale bar is 10 µm (if not indicated otherwise). Photo credit: Carlos E. Wetzel (A, D and E), Jean-Luc Biagi (B and F–I).

**Table 2 Element distribution for two example points.** Weight percentages are derived from the scanning electron analysis from two points indicated in Fig. 2F. The weight percentage of an element is the weight of that element measured in the sample divided by the weight of all elements in the sample multiplied by 100.

| Element | EDS 1 (%) | EDS 2 (%) |
|---------|-----------|-----------|
| C | 22.7 | 12.5 |
| N | 3.0 | |
| O | 36.2 | 24.5 |
| Mg | 0.4 | 0.8 |
| Al | 3.4 | 11.0 |
| Si | 24.9 | 26.3 |
| P | 0.4 | 0.3 |
| S | 0.4 | 0.3 |
| K | 3.2 | 6.1 |
| Ca | 1.4 | 2.9 |
| Fe | 4.0 | 5.5 |
| Ti | | 0.8 |

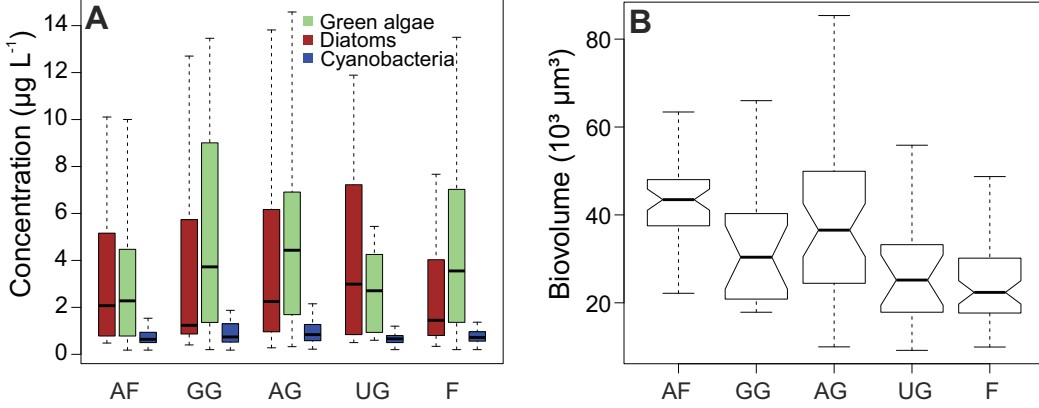

**Figure 3 Impact of habitat type on algal concentrations and diatom biovolume.** (A) Concentration of diatoms, cyanobacteria and green algae. (B) Comparison of diatom biovolumes between habitat types. Biovolume is calculated according to *Rimet & Bouchez (2012)*. AF, agricultural field; GG, grazed grassland; AG, agricultural grassland; UG, undisturbed grassland; F, forest. Not overlapping notches indicate that the medians differ (*Chambers et al., 1983*).

of the meteorological variables were included in the RDA analysis, while EC was removed. Concerning the different pigments, fucoxanthin was correlated with diadinoxanthin ($R^2 = 0.64$), while lutein was positively correlated with zeaxanthin ($R^2 = 0.59$) and with total chlorophyl a ($R^2 = 0.58$).

There was no indication of spatial autocorrelation in the pigment data according to the Mantel tests ($P > 0.05$). The total variation explained by the RDA was 40%. The first two RDA axes covered respectively 40 and 5% of the total explained variation. However, only the first axis resulted as significant ($P = 0.001$ and 0.158 respectively) after permutation test. In the 2D ordination, fucoxanthin and lutein are mainly related to the

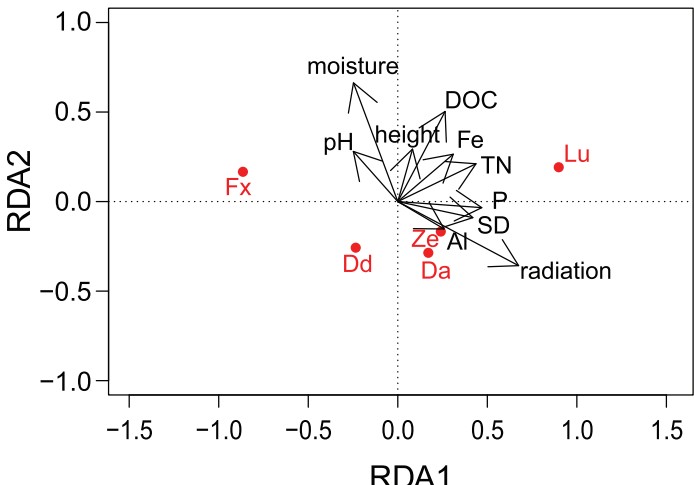

**Figure 4** **Results of redundancy analysis (RDA) on the relation between environmental variables and pigment data.** Pigment data is Hellinger transformed. Pigments: Fx, fucoxanthin; Lu, lutein; Ze, zeaxanthin; Dd, diadinoxanthin; Da, diatoxanthin. Variables: DOC, Dissolved Organic Carbon; Fe, iron; TN, Total Nitrogen; SD, Storage Deficit; P, phosphorus; Al, aluminum; radiation, global radiation; moisture, soil moisture content.

first axis, which is determined by radiation, TN and SD, while the other carotenoids follow more the second axis, which is correlated with soil moisture content (Fig. 4). Of those mentioned variables, most exhibit a strong seasonal gradient (e.g., radiation, soil moisture and SD). The Venn diagram (Fig. 5) shows that meteorological variables explain in total (including joint effects) 23.7% of the total variation ($F = 3.80$, df = 7, $P = 0.02$), whereas the site characteristic variables and soil chemical factors explain respectively 13.3 ($F = 2.88$, df = 5, $P = 0.01$) and 9.1% ($F = 2.57$, df = 11, $P = 0.027$). However, only 37.7% of the variation could be explained. Furthermore, multiple regression analysis revealed that diatom productivity (log-transformed) has a strong seasonal gradient with radiation, SD, and vegetation height all significant (ANOVA, $F > 8.73$, $P < 0.05$), whereas pH is an important factor ($F = 31.49$, $P < 0.001$) in creating a spatial gradient. Besides, diatom productivity is positively correlated with pH and vegetation height, whereas a negative relation with radiation and SD exist.

## Spatial and temporal variability pigment concentrations and diatom biovolume

Diatom abundance shows a strong seasonal variability with a peak in February and March (Fig. 6B), which occurred after the 2 wettest months and coincides with the coldest period of the sampling campaign. In the subsequent spring and summer period (April–September), abundances are low with an average of 3.9 µg L$^{-1}$ (±3.8 µg L$^{-1}$) per month. A slight increase is again visible in November. In contrast to the diatoms, green algae show less seasonality and their concentrations remain relatively high throughout the whole campaign. Furthermore, chlorophyl a reaches its highest concentrations in February and March following the pattern of fucoxanthin (Fig. 6C). Also, a peak occurred in June reaching on average 24.0 µg L$^{-1}$ while experiencing a precipitation of 90.8 mm that

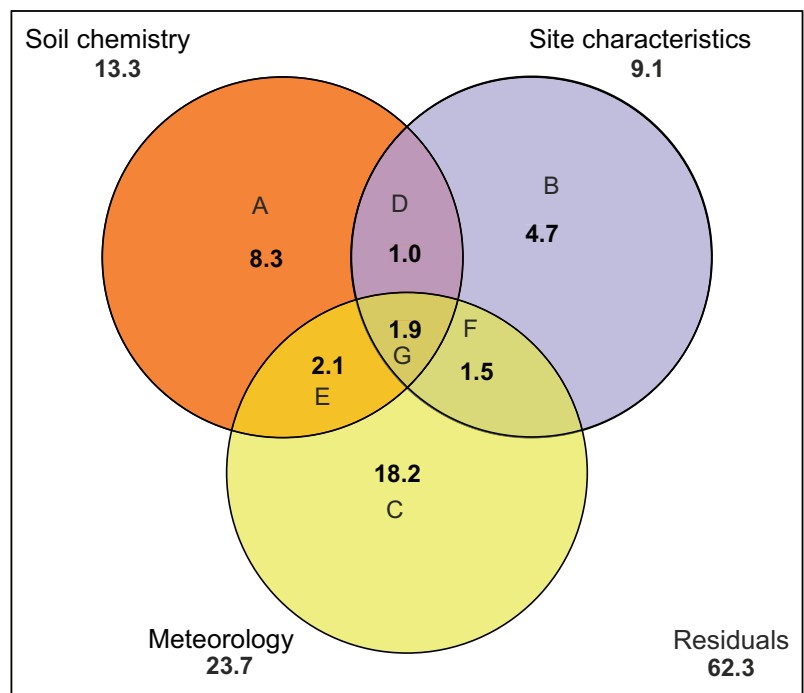

**Figure 5 Variation partitioning of the pigment composition data.** Graph is represented as a Venn diagram, indicating three groups of variables (soil chemistry, site characteristics and meteorology) with their percentages of explained variance. The diagram shows the partitioning of the explained variance into seven components. (A) Partial effects of the soil chemistry; (B) partial effects of the site characteristics; (C) partial effects of meteorology; (D) partial joint effects of soil chemistry and site characteristics; (E) partial joint effects of soil chemistry and meteorology; (F) partial joint effects of site characteristics and meteorology; (G) partial joint effects of the three groups. Numbers outside the circles stand for the total variance (in percent) explained by each variable (including all joint effects). Meteorology includes air temperature, number of frozen days, precipitation, radiation, relative humidity, wind speed and PET. Soil chemistry includes Na, Mn, Fe, pH, Si, P, DOC, TN, K, Al, Fe and Na. Site characteristics includes vegetation height, soil moisture content, site, type of habitat and level of disturbance.

month. The lowest average chlorophyl a concentrations were measured at the beginning of the campaign in December and January (2.7 and 4.1 μg L$^{-1}$ respectively).

Since larger diatoms generally contain more pigment, the diatom biovolume was calculated to check if the temporal variability of the fucoxanthin concentration was influenced by their cell size. Opposed to the fucoxanthin concentration, the diatom biovolume did not change significantly during the year (ANOVA, $F = 2.29$, df = 11, $P > 0.05$) (Fig. 6C), meaning that the temporal variability in the fucoxanthin concentration is not related to the absolute size of the diatom community. Finally, in Fig. 6D the ratio between the photoprotective carotenoids in diatoms (i.e., diadinoxanthin and diatoxanthin) and the light harvesting pigment fucoxanthin is given. This ratio generally follows the pattern in radiation/temperature levels during the year. In summer, diatoms sometimes had twice the amount of photoprotective pigments than fucoxanthin during summer time, whereas in winter the average ratio was less than or equal to one in the first 3 months of the study campaign (Kruskal–Wallis, df = 11, $P < 0.001$).

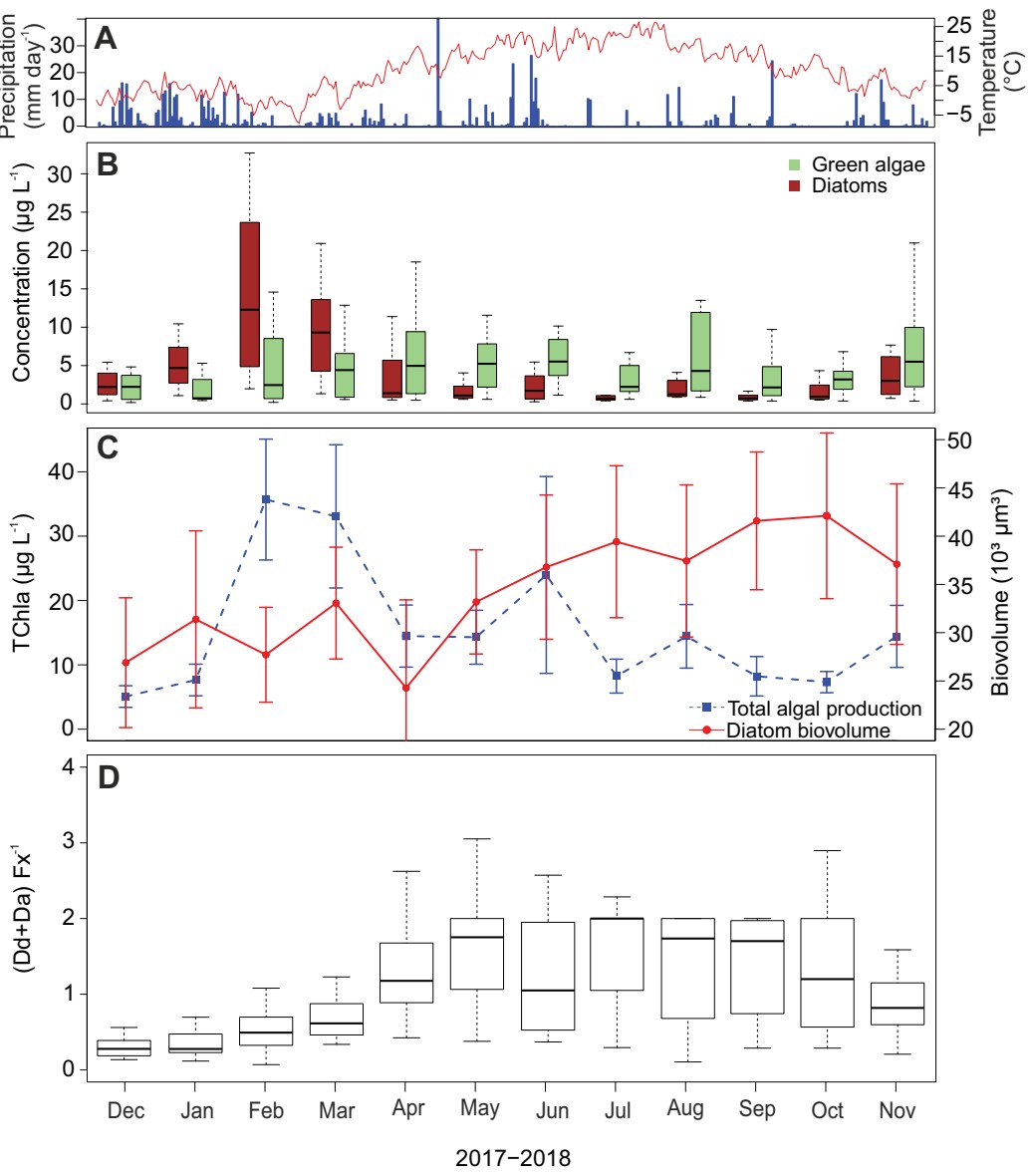

**Figure 6 Temporal dynamics of atmospheric conditions and diatom characteristics during the sampling campaign.** (A) Daily precipitation (blue bars) and air temperature (red line) fluctuations during the sampling campaign (1 December 2017–30 November 2018) retrieved from Useldange weather station (Luxembourg). (B) Temporal variation of green algal and diatom abundances. (C) Total algal production (TChla) and diatom biovolume. Diatom biovolume has been calculated according to *Rimet & Bouchez (2012)*. (D) Temporal variability of the ratio between the photoprotective (Da+Dd) and light-harvesting (Fx) pigments of diatoms.

# DISCUSSION

## Soil microbial community

The SEM image of the soil surface showed a diverse microbial community. Both diatoms and (cyano)bacteria were present. Surprisingly, green algae seemed absent. Generally, green algae are an important part of the soil algal community (*Starks, Shubert & Trainor, 1981*; *Rindi, 2011*), which our pigment data also confirms. However, for that agricultural

grassland site our data indicated that usually diatoms are more abundant than green algae during the period the sample was taken (i.e., November). Besides, species of the algal classes Xanthophyta, Rodophyta and Euglenophyta could also be present on soils. However, they generally occur in low abundances. Therefore, we did not expect to see them on the picture. Despite that pigment concentrations could be a good estimate for algal abundances, the amount of each pigment per algal cell is also dependent on its size (i.e., larger cells have a higher amount of each pigment). Overall, terrestrial diatoms and green algae have similar cell sizes (*Rindi, 2011*; *Ress, 2012*), whereas cyanobacteria have on average smaller dimensions. Therefore, the pigment signal is lower for the same number of living cells. This could explain why there are (cyano)bacteria present whilst having a lower pigment concentration than green algae. Thus, for pigment analysis, size does matter and this makes it difficult to compare pigment data with algal counts.

## Spatial and temporal variability in relation to environmental factors

In this study, we analyzed 25 environmental variables and related them to the different concentrations of fucoxanthin over the entire 12-month study period. Of those 25, we found that pH, radiation, SD and height of the lower vegetation were significant in explaining 45% of the variability in diatom productivity. The relation with pH was positive, suggesting that diatoms prefer higher pH values. This is in agreement with an earlier work by *Hoffmann (1989)*, indicating that most diatoms reach their highest abundances on neutral to alkaline soils. However, our maximum soil pH extended only to 8.11, meaning that we were unable to conclude on an optimum occurring or not. Regarding radiation and SD, both variables can serve as a proxy for soil moisture availability. The negative relation between those factors and fucoxanthin concentration was therefore expected since water, which is often a limiting factor, is essential for diatom survival and reproduction (*Camburn, 1982*; *Van de Vijver, Ledeganck & Beyens, 2002*). We also noticed that SD was a better factor for explaining the temporal and spatial variability in the concentration of fucoxanthin than precipitation and soil moisture content. This is probably because it takes both variables into account. In this case, soil moisture was a point measurement and does not give any information of the antecedent conditions, while precipitation (average of seven days prior to sampling day) does not inform on how much water is still prevailing in the upper soil layers. Finally, we found a positive relation between vegetation height and diatom abundances. This could indicate that the capacity of a higher vegetation to better keep moisture and protect against UV-radiation is important for diatom growth— particularly in warm and dry conditions (*Ress, 2012*; *Zhang, Lv & Pan, 2013*). Despite having analyzed many different environmental factors, we were still not able to explain a large part of the variation.

We observed maxima in the chlorophyl a concentration in winter and in June, following month(s) of high precipitation. *Stokes (1940)* reported the same observation and considered the antecedent moisture conditions as responsible for the high abundances in summer. Regarding the peak during winter, he considered snow cover as being important for algal growth and conjectured that a blanket of snow keeps the soil moist and

warmer than the surrounding atmosphere—creating favorable conditions for algal development. However, the results of *Davey (1991)* and *Grondin & Johansen (1995)* refute the latter, since the disappearance of snow led instead to a steep increase in the algal concentration. Although we encountered snowfall during February and March, the snow layer did not prevail for long. Thus, our winter peak could be rather the result of changes between the presence and absence of snow cover, so that soil algae at one point in time are protected against harsh winter conditions and at another point, they are able to maintain photosynthesis. An additional reason for the winter peak could be that under those low light conditions soil algae maximizes their chlorophyl a content to optimize their photosynthetic activity and decrease their chlorophyl content under high-irradiance conditions in summer (*Neidhardt et al., 1998*; *Quesada & Vincent, 1993*; *Bohne & Linden, 2002*). Indeed, according to *Kuczynska, Jemiola-Rzeminska & Strzalka (2015)* those changes are usually fast since they are engaged in basic processes (e.g., photosynthesis and photoprotection), which are essential for cell life. Thus, seasonality in chlorophyl a concentration is probably the result of the algal growth in combination with their physiological state. In addition, chlorophyl a comprises different algal groups which could respond in a different way to varying environments.

Indeed, diatoms and green algae reacted differently to seasonal conditions. While diatoms exhibited very high abundances in winter and lower abundances during summer, the abundances of green algae remained high throughout the sampling period. This indicates that green algae are more tolerant to dry and warm environmental conditions than diatoms. The reasons for this could lie in the mucilage production and the cell aggregation ability (e.g., forming filaments) of soil green algae. It is known that for green algae the extracellular matrix is well resistant against periods of desiccation and could contain a considerable amount of moisture (up to 97% of the total weight) (*Boney, 1980*; *Shephard, 1987*; *Rindi, 2011*). Furthermore, by forming filaments, they support a high self-protection against water loss (*Karsten & Holzinger, 2014*). Although many soil diatom species have developed adaptations to fluctuating moisture availabilities in terrestrial habitats (including the production of a mucopolysacharide matrix), the adaptations made by terrestrial green algae seem to be more efficient (*Shephard, 1987*; *Ress, 2012*).

Despite the fact that diatom abundances are strongly related to moisture availability, the latter obviously does not affect the diversity and composition of their assemblages. *Johansen (1993)* observed high diatom diversity in arid environments, whereas, in a related study by *Foets et al. (2020)*, diatom composition and diversity did not change significantly during the year. In addition, the latter also showed that diatom communities have a strong spatial component and are controlled by the amount of anthropic disturbance with larger species often being more tolerant (cf. *H. amphioxys*, *Pinnularia borealis* Ehrenberg). Such observations were also visible in the results of our biovolume calculations. However, there were no spatial differences in fucoxanthin concentrations, meaning that overall less living diatom cells are present in disturbed compared to undisturbed areas. On the contrary, the temporal variation in the measured fucoxanthin concentration is not influenced by the diatom biovolume, since the results of the biovolume did not show any temporal variation. Additionally, like chlorophyl, fucoxanthin has an important

function in light-harvesting and its concentration is therefore affected in the same way as chlorophyl by different irradiance levels (*Xia et al., 2013*; *Kuczynska, Jemiola-Rzeminska & Strzalka, 2015*). Because of that, we would expect the differences in the number of living diatom cells between the summer and winter period to be smaller than what we would expect from the results derived from the pigment analysis. Besides the fact that the size and physiological state of diatoms are important for pigment concentration, we could conclude that the distribution of diatom communities is mainly controlled by spatial factors related to the amount of disturbance and that diatom abundances are generally linked to temporal factors related to (soil) moisture availability.

## Recommendations and perspectives

Pigment analyses proved to be a useful tool in identifying microbial communities and deriving their abundances (*Leavitt & Hodgson, 2001*). Here, we applied the same algal extraction method as described in *Barragán, Wetzel & Ector (2018)* for diatom community analysis. However, it is recommended by the authors that only samples of bare soil or with only low presence of grass or other vegetation are taken. This is because dense vegetation does not allow a proper rinsing of the soil substrate and subsequently hinders the detachment of algal communities. As we followed this recommendation, we believe that we equally extracted the different algal groups. Furthermore, we used average diatom biovolume measurements taken from aquatic specimens (including terrestrial species occurring in those habitats). However, previous studies of *Van de Vijver & Beyens (1997)*, *Ress (2012)* and *Stanek-Tarkowska et al. (2013)* all reported that cell sizes could be significantly different (both larger and smaller) between terrestrial and aquatic habitats. We did not incorporate this, but as those changes are not unidirectional, we believe that there would only be a minimal effect on some of the results. Besides, we were not able to compare our pigment concentrations with other studies, since many different preparation, extraction and analytical methods are used and subsequently pigment concentrations are expressed in different units (e.g., $\mu g\ L^{-1}$, $mg\ cm^{-2}$, $mg\ g^{-1}$ soil) (*Leavitt & Hodgson, 2001*; *Cartaxana & Brotas, 2003*; *Schagerl & Künzl, 2007*; *Kuczynska, Jemiola-Rzeminska & Strzalka, 2015*).

Moreover, only few studies exist on the ecology of terrestrial diatoms compared to aquatic diatoms. This is rather surprising, since terrestrial diatoms and other algae are arguably affected more directly than aquatic algae by climatic changes and can be expected to respond in an immediate way (*Rindi, 2011*; *Ress, 2012*; *Souffreau et al., 2013*). Therefore, future research should give more attention to terrestrial microbial communities, certainly in the light of climate change.

## CONCLUSIONS

In this study, we investigated the temporal and spatial variability of soil diatom abundances and compared them with the abundances of green algae and cyanobacteria. The abundances were derived from pigment analysis using HPLC. Our results supported our first hypothesis that diatom abundances show seasonal succession and that their

abundances are mainly controlled by factors influencing the (soil) moisture availability (e.g., radiation and storage deficit). However, our second hypothesis was rejected, since the effect of habitat type, which played a key role in shaping the diatom communities (*Foets et al., 2020*) and subsequently their biovolume, was not seen in the pigment concentrations. This led to the conclusion that overall less diatom cells are present in disturbed than in undisturbed habitats. Contrary to diatoms, green algal productivity remained stable over the course of the study period. Also, we observed that diatoms could have higher abundances than green algae and cyanobacteria at periods of the year with high soil moisture. Concerning future studies, more focus should be on the ecology of terrestrial diatoms and other algae, since they are affected more directly than aquatic algae by climatological conditions (*Rindi, 2011*; *Hinder et al., 2012*).

## ACKNOWLEDGEMENTS

The authors are very grateful to Emmanuelle Cocco for the pigment analysis and developing the protocol. They also thank Dr. Denis Pittois and Johanna Ziebel for analyzing the soil nutrients. Three reviewers are thanked for their constructive comments that significantly improved the manuscript.

### Funding

This work was supported by the Luxembourg National Research Fund (FNR) (PRIDE15/10623093/HYDRO-CSI). The funders had no role in study design, data collection and analysis, decision to publish, or preparation of the manuscript.

### Grant Disclosures

The following grant information was disclosed by the authors:
Luxembourg National Research Fund (FNR): PRIDE15/10623093/HYDRO-CSI.

### Competing Interests

The authors declare that they have no competing interests.

### Author Contributions

- Jasper Foets conceived and designed the experiments, performed the experiments, analyzed the data, prepared figures and/or tables, authored or reviewed drafts of the paper, and approved the final draft.
- Carlos E. Wetzel conceived and designed the experiments, authored or reviewed drafts of the paper, and approved the final draft.
- Adriaan J. Teuling conceived and designed the experiments, authored or reviewed drafts of the paper, and approved the final draft.
- Laurent Pfister conceived and designed the experiments, authored or reviewed drafts of the paper, and approved the final draft.

## Data Availability

The data and an explanation of the data are available in the Supplemental Files.

Diatom community data from which the diatom biovolume has been calculated is available at Mendeley: Foets, Jasper (2019), "Temporal and spatial variability of terrestrial diatoms at the catchment scale", Mendeley Data, v2 DOI 10.17632/9s28gvnr53.2.

## Supplemental Information

Supplemental information for this article can be found online at http://dx.doi.org/10.7717/peerj.9198#supplemental-information.

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
