# Peer review of "Temporal and spatial variability of terrestrial diatoms at the catchment scale: controls on productivity and comparison with other soil algae"

_PeerJ, doi:10.7717/peerj.9198_

## Round 0.1 · original submission · Major Revisions

Please, address carefully all the reviewers comments and resubmit within the next 55 days.

·

Basic reporting

Manuscript #45133 “Temporal and spatial variability of terrestrial diatoms at the catchment scale: controls on productivity and relation to other soil algae” by Jasper Foets, Carlos E. Wetzel, Adriaan J. Teuling and Laurent Pfister – Submitted to PeerJ. Review report

The authors assessed the variability of terrestrial diatoms over a catchment in Luxembourg, over space (16 sites) and time by conducting monthly sample collections over 1 yr. Diatom species and pigment concentrations were analyzed and correlated with sites characteristics. Based on the references cited in the introduction, it is interesting to see that this subject has not been tackled recently, although it is topical in a context of global change. From the title (“relations to other soil algae”), I would have, however, expected statistical analyses and in-depth discussion on the relationships between algal groups under variable sites and dates, and maybe on the competitive abilities of such organisms in soil environments.

The paper is generally well written. I am not a native speaker but I found it easy-to-read. The abstract provides a comprehensive overview of the study and the obtained results. In the introduction, the authors provide important background information. However, the identification of research gaps is somewhat lacking.

Experimental design

The methods section gives an overview on the analyses made. The study design and the chosen endpoints are relevant. I found some inconsistency in presenting the “Scanning electron microscopy analysis” before having described the sample collection. Consequently in that paragraph, I missed some important information on which samples were analyzed, what the authors meant by “non disturbed soil samples”. Are they the cores described later on (line 150)? Were they fixed? Frozen? Lyophilized? Coated?

Some more details could also be provided in order to be able to reproduce the work. Important aspects I came across are: an overview of the sites’ characteristics could be added (in a summary table within the main manuscript, complementary to the detailed dataset provided in supplementary material), the lack of information on reference substance tests and of description of validity criteria (which are the certified standards you used and what are the detection limits for them?). Is there a reason why you chose Fe and Al as specific metals for analysis (line 181), I this that this should be justified/explained by the authors here. Then you specify that the pH was measured, but was it used to calculate/estimate the free metal ions concentrations? Finally, I have some questions about the statistical evaluation: on which data is the DCA performed? On diatom community composition (as you mention “species turnover”)? This is unclear to me. For diatom biovolume calculations, did you use all diatom species found in the samples, or did you remove rare species below a threshold of relative abundances? In this case please detail.

Validity of the findings

The results section provides a good overview on the results, but I was surprised by the fact that they are described in the present tense, instead of the commonly used simple past. I have a specific reservation on the description of the results made on line 301-302. I think there is an incorrect formulation of the results you observed, linked to the fact that some variables are correlated and of them, some explaining ones are removed. More precisely what I mean here is that I strongly disagree with the following statement “diatoms sometimes had twice the amount of photoprotective pigments […] when the temperature is high” suggesting that temperature is the driving force for this increase in photoprotective pigments. I would rather think that protoprotective pigments are overproduced under increasing incident light (also see what you discuss on lines 336-338 about the photoprotection from incident UVs by higher vegetation height). Parameter which is obviously potentially correlated with higher temperature. I suggest rephrasing this sentence.

The discussion section is compiled following similar organization as in the results. The findings are compared with relevant literature in the field and the section provides a critical assessment of the results and suggestions for further research. I have a specific comment on the biovolume calculations and conclusions drawn (lines 379-384). The fact that you found no correlations can be explained by the methodology used. Indeed, diatom biovolumes were calculated here from literature data, which are an average of the species dimensions published. It is well-known that there is a large range of variability in diatom cell sizes, that may be related to their specific habitat conditions. Maybe they were missed here, and would have been seen using measured dimensions. Could you discuss on this?

The supplementary material provides valuable additional information on the sites characteristics and pigment analyses.

Additional comments

Overall, I think this is an interesting and topical study, fitting well to the scopes of the journal. I recommend to accept it with moderate revisions.

You will find additional specific comments below.

Introduction
Line 64: Be careful with the use of “evolution”. This term usually refers to some Darwinist meaning (e.g. see Forbes, A. A. & Krimmel, B. A. (2010) Evolution Is Change in the Inherited Traits of a Population through Successive Generations. Nature Education Knowledge 3(10):6), maybe prefer a more unspecific word such as “change”?
Line 111: Remove capital to “Soil”
Line 111: How is defined “diatom productivity”? Could you specify here?

Materials and Methods
Line 150: Could you specify the height of the cores collected?

Results
Line 250: “whereas the concentrations of zeaxanthin were significant for both” is unclear to me? Do you mean both sampling campaigns and whole dataset?

Discussion
Line 310: Not sure that “certainly” is needed here
Lines 311-312: Could you provide any reference to support this statement?

Legends
Table 1
Change “,” for “;” after “(diatoms)”
Figure 1
Please italicize “Nitzschia pusilla”
Figure 3: All abbreviations in the RDA should be explained in the legend to make the reading of illustrations (tables and figures) easier.

Reviewer 2 ·

Basic reporting

Review on “Temporal and spatial variability of terrestrial diatoms at the catchment scale: controls on productivity and relation to other soil algae (#45133)” by Foets et al.

This paper present novel data on he structure and physiology of soil algae, specially focused on diatom ecology which is one of the fields of expertise of the authors. I have no serious objections against the publication of the paper. Hereby some comments that may aid the authors to improve the draft.

Experimental design

- I do not see the point of comparing biotic and abiotic variables measured simultaneously. I mean, the algal assemblage observed in soil samples has grown there for some weeks, and its characteristics (cell numbers, pigments…) reflect the environmental conditions during that period. However, only the data measured during sampling is taken into account. Since the authors have temporal data, maybe it is worth to use “time-weighted averages” or similar.
- Some data about the occurrence of dead, empty (allochtonous?) frustules would be useful.
- The reader is referred to a previous paper by the same authors with additional data on methods and sampling design. I’d appreciate rather to see all the data here, even if implies repetition of published info. I am particularly interested in the spatial arrangement of samples. In this regard, the statistical tests used depend strongly on whether the samples are independent or not. If the authors considered only a single “experimental unit” (the Attert River basin in Luxembourg) the samples can hardly be considered independent between them. This would require the use of “repeated measurements” tests rather than the ones actually employed. Spatial autocorrelation is neither considered. All this puts at risk the plausibility of the conclusions rendered.
- Specific concerns on statistics: Pearson’s test is used without previously testing its assumptions. It is said that correlated variables were removed prior to further analysis: both variables were removed?
- It is not stated whether if log transformation actually accomplished normalization. By the way, about data transformation: many biometricians rely on nonlinear transformations (logarithmic, square root, or reciprocal) because they jointly affect the linear predictor of the average as well as the distribution of the random component, and the joint effect may not always be that which is sought (Gregorie et al. 2008). Model parameters will lose their biological meaning along with distorting the functional relationship as a whole (Onofri et al. 2010). Thus, the transformed data do not necessarily bear any information of the original data, and the consequence of data transformation is not predictable in general (Norleans 2000). See also Games (1984).
- It should be highlighted somewhere (and not only in the figure) that residuals (unexplained) accounts for most (60%) of the variance in the dataset.
- A pity that the authors did not use their own cell measurements but that of Omnidia (averages). It has been reported that soil populations may have different morphometry than aquatic conspecifics.
- Kruskal-Wallis test assumes a similar shape in the distributions, except for a possible difference in the population medians. This is not demonstrated in the paper.
- Some name misspellings: pussila.

Validity of the findings

- The text is well documented and discussed, with updated references and, in general a good scientific style and English language.
- The approach followed by the authors is original and fills a gap in our knowledge of soil microbiology
- The conclusions are well supported by the findings reported. The figures and tables are OK

However, see above

Additional comments

Having all this into account, I cannot recommend the publication of the paper until all these issues are adequately corrected or discussed.




References

Games, P. A. (1984). Data transformations, power, and skew: A rebuttal to Levine and Dunlap.
Gregoire, T. G., Lin, Q. F., Boudreau, J., & Nelson, R. (2008). Regression estimation following the square-root transformation of the response. Forest Science, 54(6), 597-606.
Norleans, M. X. (2000). Statistical methods for clinical trials. CRC press.
Onofri A, Carbonell E.A, Piepho H.P, Mortimer A.M, Cousens R.D. 2010. Current statistical issues in Weed Research. Weed Res. 50: 5- 24. 4.

Reviewer 3 ·

Basic reporting

no comment

Experimental design

no comment

Validity of the findings

no comment

Additional comments

I have some minor comments and suggestions for improvements of the manuscript:

i) METHODS: It is not clear for me, why the authors log-transformed abiotic parameters except for pH prior to ordination analysis. I understand this approach in a biplot that shows the relationship between pH values and other type of data. However, to my knowledge, standardization (i.e., rescaling all the data of different range), not log-transformation, is usually performed prior multivariate analysis such as RDA.
ii) RESULTS AND FIGURE 4: It would be helpful to know the results of the tests for variation partitioning approach. I would suggest adding the P-values of the tests to the results and to the Venn diagram.
iii) DISCUSSION: I would appreciate some discussion about the results of SEM chemical mapping.
iv) INTRODUCTION AND/OR DISCUSSION: The authors mentioned three groups of algae present in temperate soils, cyanobacteria, green algae, and diatoms. Is it possible that Eustigmatophytes and Xanthophytes were also present in their samples? How common and abundant are these groups of Ochrophytes in temperate soils?
v) DISCUSSION: The authors discussed the importance of cell size on chlorophyll concentrations between cyanobacteria and diatoms. How large are the cell size differences between green algae and diatoms?
vi) REFERENCES: “Vincent, W.F., (2000)” has incorrect page numbers: 321-240.

---

## Round 0.2 · accepted · Accept

The authors have answered adequately to the questions raised by the reviewers.

Reviewer 2 ·

Basic reporting

The authors have responded adequately to the questions raised. Hence, the paper has been substantially improved. I have now no objections to see this article published in PeerJ.

Experimental design

No comments

Validity of the findings

No comments

Additional comments

See above